# Research on the Relationships between Psychological Problems and School Bullying and Non-Suicidal Self-Injury among Rural Primary and Middle School Students in Developing Areas of China

**DOI:** 10.3390/ijerph17103371

**Published:** 2020-05-12

**Authors:** Xuyang Li, Feiyu Chen, Yixiang Lin, Zhihui Jia, Winter Tucker, Jiangyan He, Lanyue Cui, Zhaokang Yuan

**Affiliations:** 1Jiangxi Province Key Laboratory of Preventive Medicine, School of Public Health & Queen Mary School, Nanchang University, Nanchang 330006, Jiangxi, China; xuyangl0526@email.ncu.edu.cn (X.L.); linyixiang338@163.com (Y.L.); j18563867699@126.com (Z.J.); cuilanyueliang@163.com (L.C.); 2Center for Disease Control and Prevention in Dongxiang District, Fuzhou 331800, Jiangxi, China; dxaids@163.com (F.C.); dxcdcjjk@163.com (J.H.); 3School of Community Science, University of Nevada, Reno, NV 89557, USA; WinterT@nevada.unr.edu

**Keywords:** primary and middle school students, psychological problems, school bullying, non-suicidal self-injury, developing areas of China

## Abstract

(1) Purpose: To analyze the role of psychological problems in connection with school bullying and non-suicidal self-injury (NSSI) among rural primary and middle school students in developing areas of China. (2) Methods: A multi-stage, stratified, cluster random sampling method was used to select 2284 rural primary and middle school students in Jiangxi Province for study. Questionnaires regarding the health risk behaviors of children in developing areas were left behind at primary and middle schools, and they were later collected and analyzed by using the mental health diagnostic monitoring scale for Chinese primary and middle school students. Pearson correlation, logistic regression, and bootstrap tests were conducted to analyze the association between psychological problems, school bullying, and NSSI. (3) Results: The incidence of NSSI in rural primary and middle school students in Jiangxi Province was 14.84%. Compared with other children with behavioral problems, those who had experienced school bullying and had mild/severe psychological problems were more likely to have engaged in NSSI behaviors (*p* < 0.001). Psychological problems have a mediating effect between school bullying and NSSI, which accounted for 12.96% of the total effect. (4) Conclusion: Psychological problems are likely an effect modifier in the connection between school bullying and NSSI behaviors. Therefore, effectively targeting psychological problems in rural primary and middle school students in Jiangxi Province may help prevent and control NSSI behaviors in students who have experienced school bullying.

## 1. Introduction

Non-suicidal self-injury (NSSI) is defined as the direct or intentional harm of an individual’s body tissue without a suicidal intention. The common forms of this include beating, cutting, burning, and scratching [1]. This kind of behavior is socially unacceptable and has the characteristic of generally being non-lethal. NSSI is considered a widespread and growing public health problem, especially among adolescents [2]. Studies in Asia [3], Australia [4], Europe [5], and the United States [6] have found that among adolescent students, the lifetime prevalence of NSSI is between 10% and 32%. Swannell [7] conducted a systematic review of published NSSI studies worldwide and found that the lifetime prevalence of adolescent self-injury was 17.2%. A prospective cohort study in the United Kingdom showed that during the following year, the risk of suicide by NSSI adolescents was 0.7%, which is approximately 66 times that for all adolescents. Over the next 5, 10, and 15 years, the risk of suicide increases from 1.7% to 2.4% and 3%, respectively [8]. According to a Chinese youth survey, the incidence of NSSI behavior in the past 12 months was 26.1%. The proportion of rural left-behind youth who reported at least one NSSI behavior was 47.3%, which was much higher than the proportion of non-left-behind youth, and also much higher than the average level in foreign countries. Studies have shown that the overall incidence of NSSI behavior in this group has a high incidence, a long duration, and the tendency to be repeated [9].

A victim of bullying is someone who experiences repeated, offensive behavior, such as unprovoked attacks, social isolation, humiliation, ridicule, malicious rumors, and abuse, resulting in severe distress for the victim, who is unable to defend himself. Different types of bullying include language bullying, relationship bullying, and physical bullying, and the most common place where bullying occurs is in school [10]. School bullying is a significant source of social pressure on young people. A survey of 104,825 primary and middle school students in 29 counties in China in 2016 found that the incidence of school bullying in primary and secondary schools was 33.36%, which was higher than survey results in Northern Europe, Switzerland, Australia, and the United States [11]. School bullying is a global public health problem among adolescents, and it is related to many negative behavioral and emotional outcomes, including psychological disorders such as anxiety, depression, as well as loneliness, low self-esteem, social withdrawal, and poor social adaptability [12]. It has been independently linked to psychological problems from pre-existing mental health status, genetic predisposition, or family psychosocial difficulties [13]. A number of foreign meta-analyses have shown that bullying is likely the root cause of severe negative psychological outcomes, including depression, anxiety, and low self-esteem [14,15]. A survey of Chinese adolescents showed that there was also a horizontal link between bullying victims and mental health issues such as depression, anxiety, inferiority, and loneliness [16].

Bullying in childhood or adolescence is a risk factor for deliberate self-injury and suicide-related behavior at any age. Hay and Meldrum studied the impact of bullying on NSSI in adolescents in the southeastern United States and found through a series of multiple regression analyses that bullying was positively linked to NSSI [17]. Van Geel et al. conducted a meta-analysis of nine non-clinical adolescent samples and found that school bullying is one of the main reasons that adolescents turn to NSSI. Compared with children who did not suffer from school bullying, the probability of an adolescent who had been bullied turning to NSSI more than doubled, which further increased the likelihood of adolescents adopting suicide-related behaviors [18]. The relationship between bullied and NSSI can be adjusted through variables such as self-deprecation, self-criticism, and low self-esteem. Most self-injured people report the reasons for hurting themselves as being self-punishment or an expression of anger toward themselves/others [5]. In addition, school safety and school care are related to school bullying and are also important factors in the mental health of adolescents. Pupils who are bullied often attend schools with poor school safety records. Academic pressure on students and the lack of sufficient teacher support are also related to adolescent NSSI, especially among female adolescents [19]. The literature indicates that depressive and anxiety symptoms are also associated with NSSI. Among groups affected by NSSI, the prevalence of psychological disorders and anxiety symptoms were 61.2% and 57.7%, respectively. Depression symptoms at the first moment significantly predicted NSSI at 2- to 3-fold [20]. A study by Laurence using data from Belgian and Dutch high school students found that the relationship between bullying and self-injury was partially mediated by depressive symptoms [5].

Jiangxi Province is one of China’s developing regions. The regional population is highly mobile, and a large number of surplus rural laborers have entered the cities to work. As a result, a large number of school-age youth have been left at what was once the family home by one or both parents due to household registration and family economic difficulties and were therefore left behind. These youth were in the formation period of psychosocial adaptability. Due to the lack of effective parental supervision and limited educational resources, left-behind youth may not be able to properly resolve life’s difficulties or to properly be able to learn to do so and may, as a result, suffer a series of serious health risk behaviors. Compared to children who were non-left-behind (rural, school-aged youth who indicated that neither of their parents worked elsewhere in China ) in Jiangxi Province, the “left-behind” youth were more likely to suffer from a higher rate of school bullying, psychological problems, and NSSI behaviors [21,22]. Previous studies have shown that school bullying and psychological problems may have an impact on NSSI [20], and school bullying also increases the occurrence of adolescent psychological problems [12]. Whether school bullying can affect the mechanism of NSSI through psychological problems needs further research and confirmation. In this study, a sample of 2284 rural primary and middle school students in Jiangxi Province, an underdeveloped region of China, was used to better understand the factors in the relationship between school bullying and NSSI behavior, and to examine the mediating effect of psychological problems on school bullying and NSSI. The study’s results may be useful in creating effective and targeted measures to reduce the occurrence of NSSI and school bullying among rural youth and provide a basis for relevant departments to carry out the next step in mental health assessments of rural adolescents in order to formulate an intervention strategy for these youth in Jiangxi Province.

## 2. Methods

### 2.1. Participants and Procedures

In September 2018, a multi-stage stratified cluster random sampling method was adopted in Yudu County of Ganzhou City, Shangrao County of Shangrao City, Duchang County of Jiujiang City, Fengcheng County of Yichun City, Dongxiang County of Fuzhou City, and Suicchuan County of Ji’an City. A county (township level) public central elementary school and middle school were taken from each county. These sample schools were characterized by a wide distribution of students (from various natural villages within the jurisdiction), a high proportion of whom have parents working outside (data provided by the local education department). The age range of the respondents was between 7 and 16 years old, completely covering the primary and secondary school stages. Self-reported questionnaires were administered to primary and secondary school students in the counties. Questionnaire 1 refers to the US Centers for Disease Control and Prevention (CDC) Youth Risk Behavior Surveillance System (YRBSS) questionnaire which was translated into Chinese and re-designated the “Questionnaire for Health Risk Behavior of Left-Behind Children in Rural Areas in Less Developed Areas”. The factorial analysis results showed that the Kaiser–Meyer–Olkin measure of sampling adequacy was 0.704, and Bartlett’s test of sphericity was χ^2^ = 7270.52, d_f_ = 136 (*p* < 0.001), suggesting that it was more suitable for factorial analysis. The Cronbach α coefficient of the questionnaire was 0.76. Questionnaire 2 was revised by Professor Zhou Bucheng of the Department of Psychology of East China Normal University and other psychology researchers based on the “Diagnosis Test for Disturbance Disorder” prepared by Kiyoshi Suzuki and others to adapt it to the Mental Health Diagnostic Test Scale (MHT) standardized by Chinese primary and secondary school student tests, good reliability, and half-scale reliability of 0.91.

A total of 2435 questionnaires were distributed. Logical screening and deletion of invalid records were performed by SAS 9.4 software (if a participant had not completed more than 30% of a measure, they were classified as incomplete, and their data were omitted from the analysis of the scale). The survey returned 2284 valid questionnaires, with an effective recovery rate of 93.80%. For these, there were 1208 boys (52.89%), 1076 girls (47.11%); 1348 primary school students (59.02%); 936 middle school students (40.98%); left-behind children accounted for 26.71%.

### 2.2. Measures

**School bullying.** Victims of school bullying in this study were individuals who had been repeatedly teased, threatened, or had rumors spread about them, or beaten, shoved, or hurt by one or more students. If both parties were of equal strength, or there was mutual teasing between friends, such was not counted as bullying. The “Yes” or “No” option was used to judge whether the student has suffered school bullying [23].

**Psychological problems.** The Psychological Diagnostic Test Scale (MHT) for elementary and middle school students was used to understand symptoms currently experienced by the study subjects, including the three dimensions of anxiety, bad emotional tendency, and bad physical symptoms. There are 100 entries in the subscale. If the answer is “yes”, the respondent will be assigned 1 point. If the answer is “No”, the respondents will not be assigned any points. Excluding the effectiveness scale items, the scores of all remaining questionnaire items are added up to obtain the full-scale points. The respondent whose score is larger than 8 will be classified as “severe problems”, 3–7 points as “mild problems”, and less than 3 points as “normal” [24,25].

**Non-suicidal self-injury.** It was defined as behaviors where the respondent had not suicidally hurt themselves in the past 12 months, including 9 types, which are self-inflicted knife or sharp weapon injuries, scalding caused from cigarettes, scratching or pulling out of hair, hitting one’s head against a wall, biting hard on oneself, tying up one’s body with rope, eating inedible objects, and immersing oneself in water so as to drown. The total number of occurrences of various self-injurious behaviors was counted as the frequency of self-injury. Five grade points were used: 0 times as “none”, 1 time as “mild”, 2 times as “moderate”, 3 times as “severe”, 4 times and above as “extremely heavy”, counting 0 to 4 points, respectively, which is the NSSI occurrence score [26].

**Covariates.** We included 9 potential confounding variables from the standardized YRBSS: household domestic violence, physical abuse by an adult, academic record, and academic pressure. Academic pressure was assessed by asking the students the following: “What level of stress do you feel as a result of your studies?”. Ordinal response items were dichotomized as high academic pressure (very stressed, above-average stress) vs. median/low academic pressure (average stress, below-average stress, no stress). The other two questions were dichotomized as yes versus no.

Demographic covariates included sex, phase of studying, left-behind children, being an only child, and family structure. Left-behind children, according to the State Council’s Opinions on Strengthening the Care and Protection of Rural Left-behind Children, are defined as children younger than 16 years old, of whom both parents have been away for more than 6 months, or one child who has been out for more than 6 months, and the other party does not have the ability to support and monitor. The only child in one’s family was assessed by asking participants: “Are you the only child of your parents?” (Yes or No). Finally, students were asked a question about family structure: “What is your family’s structure?”. Ordinal response items were dichotomized as a two-parent family vs. non-parental family (single-parent family, reconstituted family, and other).

### 2.3. Data Analysis

Epidata3.0 was used for data entry, and SPSS 22.0 software was used for data analysis and processing. Pearson chi-square test was used to compare the relationship between demographic characteristics such as gender, school period, left-behind children, only child, and family structure, and NSSI without adjustment.

Binary multi-factor logistic regression analysis was used to explore the relationship between school bullying, psychological problems, and non-suicidal self-injury. At the same time, covariates were controlled to calculate the adjusted odds ratio (AOR) and 95% confidence interval (95% CI).

The correlation between bullying, psychological problems, and NSSI was explored using Pearson correlation, and exploratory factor analysis was used to conduct a common method bias test on the data. Amos 22.0 was used to construct a structural equation model (SEM), and a bootstrap method was used to test the mediation effect. The test level was α = 0.05.

### 2.4. Ethical Aspects

Approval was obtained from the principals of the schools participating in the survey. Class teachers administered the survey questionnaire survey during a regular school day. Parental opt-out consent to participate in the survey was obtained in advance. Students were informed that participation was anonymous and voluntary. If they did not want to participate in the survey, this period of time could be used to do their own classwork, provided that the classmates who fill out the questionnaire would not be affected. The demographic information did not allow the identification of individual students involved in the survey. The protocol was approved by the Ethics Committee of the second affiliated hospital of Nanchang University.

## 3. Results

### 3.1. Demographic Distribution of NSSI Behavior among Rural Primary and Middle School Students in Jiangxi Province

Among the 2284 respondents, a total of 339 students in rural areas in Jiangxi Province have been detected NSSI behavior, with an incidence rate of 14.84%. Among those, the incidence of students in non-two-parent families (20.26%) was higher than that in two-parent families (13.76%). Witnessing household domestic violence (17.25%) was higher than that of students who did not witness it (13.68%), physical abuse (15.44%) was higher than those who did not experience any(6.92%), and those with high academic pressure (18.57%) was higher than that of those with medium/low (12.63%) pressure. Those who had been bullied (24.11%) were higher than those who did not suffer bullying (10.49%), and those with mild/severe psychological problems (19.92%) were higher than normal (0.17%). All these results were statistically significant. See Table 1 for details.

### 3.2. Testing of Common Deviation Method

The Harman single factor test was used to test for common method bias. The results of unrotated principal component factor analysis showed that there were 12 factors with characteristic root values greater than 1, and the variation value explained by the first factor was 21.48%, which is less than the critical criterion of 40%, indicating that the common method of the study is not serious.

### 3.3. Multivariate Logistic Regression Analysis

Based on whether rural primary and middle school students (0 times) or not (≥1 time) have taken NSSI behaviors as the dependent variables, We used gender (0 = male, 1 = female), phase of studying (0 = primary school, 1 = middle school) ), left-behind children (0 = Yes, 1 = No), only a child in one’s family (0 = Yes, 1 = No), family structure (0 = two-parent families, 1 = non-two-parent families), household domestic violence(0 = Yes, 1 = No), physical abuse (0 = Yes, 1 = No), academic performance (0 = High, 1 = Medium/Low), academic pressure (0 = High, 1 = Medium/Low), school bullying (0 = Yes, 1 = No), psychological problems (0 = Normal, 1 = Mild/Severe problems) as the independent variables, and included the multi-factor non-conditional binary logistic regression model, and the inclusion standard α = 0.05.

Results of the analysis of the results showed that rural non-parent family students (AOR = 1.29; 95% CI = 1.00, 1.66) in Jiangxi Province had a higher tendency of NSSI than children of two-parent families (*p* < 0.05). Having experienced physical abuse (AOR = 2.52; 95% CI = 1.33, 4.76) compared with those without this behavior, they had a higher tendency of NSSI (*p* < 0.01). After controlling for covariates, students who had suffered school bullying (AOR = 2.10; 95% CI = (1.65, 2.69)) had a higher tendency to NSSI than students who had not suffered (*p* < 0.001). According to the results of the self-assessment psychological survey, those with mild/severe psychological problems (AOR = 125.03; 95% CI = 17.49–893.89) had a higher incidence of NSSI than normal students (*p* < 0.05). See Table 2 for details.

### 3.4. Correlation Analysis between NSSI and Various Factors of School Bullying and Psychological Problems in Rural Primary and Middle School Students

As shown in Table 3, correlation analysis found that whether NSSI occurred in primary and middle school students was positively correlated with anxiety, bad emotional tendency, bad physical symptoms, physical bullying, verbal bullying, and relationship bullying. The more likely each factor is to occur, the more likely these students are to have NSSI.

### 3.5. One-Factor Mediating Effect of Psychological Problems between School Bullying and NSSI

Based on the results of correlation analysis, we performed a confirmatory factor analysis on three latent variables suffering from school bullying and psychological problems in the model and measured physical bullying, verbal bullying, and relationship bullying from three dimensions. Anxiety, bad emotional tendencies, and bad physical symptoms are important indicators to measure the psychological problems of primary and middle school students. There is a positive correlation between school bullying and psychological problems (r = 0.022, *p* < 0.001).

This paper explored the one-factor mediating effect of psychological problems by using school bullying as a predictive variable and NSSI as an outcome variable. The model fitting indices of the intermediary model MO constructed by the structural equation model are: χ^2^/d_f_ = 2.328, CFI = 0.997, TLI = 0.994, GFI = 0.996, RMSEA = 0.024, SRMR = 0.013; see Figure 1. We construct an SEM to explore the mechanism of school bullying and psychological problems on NSSI in rural primary and middle school students in Jiangxi Province and test the hypothesis model, which includes the following hypotheses: School bullying in primary and middle school students has a direct effect on psychological problems and NSSI; psychological problems have a direct effect on NSSI; suffering from school bullying has an indirect effect on NSSI through psychological problems. In the end, the independent variable was suffering from school bullying, and the mediating variable is a psychological problem. It acted on the structural equation model of NSSI. The fitting index shows that the model fits well, as shown in Figure 1.

The direct and indirect effects in the model were tested using the bias-corrected non-parametric percentage bootstrap method. The direct and indirect effects in the model were tested. The bootstrap method was used. After repeated sampling 5000 times, the 95% confidence interval was calculated. The results showed that 95% of the total effects and the mediating effects of psychological problems did not include 0, indicating that the mediating role of bad psychology in school bullying and NSSI was fully confirmed. Among them, the mediating effect of psychological problems between the two was 12.96%. Therefore, the one-factor mediation model established in this study has statistical significance; see Table 4.

## 4. Discussion

This study focused on rural primary and middle school students in developing areas of China. It mainly revealed the relationship between school bullying and psychological problems of non-suicidal self-injury behaviors after controlling for factors such as demographic characteristics. It also expanded upon the research on this aspect of rural primary and middle school students in developing areas of China. Consistent with previous research, primary and middle school students who have suffered school bullying and have psychological problems have a higher risk of NSSI [22,27,28]. In addition, the bootstrap test of this study showed that the psychological problems of rural primary and middle school students in developing areas of China play a mediating role between school bullying and NSSI.

The results of this survey show that the incidence of NSSI in rural primary and middle school students in China’s developing areas is 14.84%, which is higher than the previous survey results of Chinese students [29,30], indicating that the NSSI problem in rural primary and middle school students in China’s developing areas cannot be ignored. The results of multi-factor logistic regression analysis showed that non-parent families and students who had experienced physical abuse were risk factors for NSSI, which is consistent with the results of previous studies [22,31]. Studies have shown that the integrity of the family structure and the importance of family education play a crucial role in the growth of primary and middle school students. On the one hand, one or both parents have not been around for a long time, and the children are in the state of intergenerational fostering so that ordinary family functioning was lacking, which led to defects in the children’s personality development [32]. On the other hand, students who had experienced domestic violence may have caused them to bear a greater degree of stress during their growth period. This chronic source of stress may have affected the biological ability of adolescents to regulate stress. In this stage, emotional instability, anxiety, and psychological conflict often occur. In order to alleviate this conflict, the youths often participate in some alternative behaviors, such as self-injury, to regulate painful emotions and receive psychological release [33,34].

After controlling for covariates, having suffered school bullying and having mild or severe psychological problems were assessed as risk factors for students to turn to NSSI. The dimensions of school bullying were positively related to the dimensions of NSSI and psychological problems. Further analysis using structural equation models found that school bullying was predicting psychological problems and NSSI positively. It showed that school bullying, as a kind of bad school behavior, was one of the most common sources of interpersonal stress during adolescence, and may be related to NSSI and bad psychology of rural primary and middle school students in under-developed areas of Jiangxi Province, which is consistent with the results of previous cross-sectional studies and one of two longitudinal studies [18,35], but contradicted another longitudinal study that found no significant predictive effect of peer bullying on self-injury [36]. There may be two reasons for this inconsistency. The first reason may be related to the assessment method. For example, in the study by Heilbron and Prinstein, bullying was measured by peer nomination, but bullying in other literature was measured by self-reporting. The second reason may be related to the follow-up time of the study. The Heilbron and Prinstein study followed the participants for two years, while other studies have conducted cross-sectional surveys of adolescents. Peer bullying may affect self-injury within a relatively short period of time [35,36]. The harm caused by school bullying to youths’ physical and mental health cannot be ignored. It will not only lead to a significant decline in the students’ academic performance and problems in academic adaptation, and may also reduce their sense of self-worth and more easily lead to emotional depression, which contributes to negative personality, directly affecting the normal development of its social nature [37]. Previous research results showed that the effects of being bullied during the school years may last until older adulthood, and those who have suffered bullying are bullied are more likely to have emotional problems, such as anxiety, inferiority, and self-harm and suicide risk [38]. A qualitative study by Long et al. showed that many of their respondents used self-injury as a means to manage traumatic life experiences and emotional distress. From this perspective, self-injury has become a bad coping mechanism by those trying to cope with having been bullied in the past [39]. In addition, previous research has also found that bullying makes adolescents face more problems in terms of peer relationships, school disengagement, and mental health. Our research also showed that school bullying is associated with NSSI. Therefore, school bullying should be regarded as an important red flag in developing rural areas of China, and intensive interventions are warranted to improve the situation or prevent them from occurring [19].

The dimensions of psychological problems are also positively correlated with NSSI, and they are also suggested in the structural equation that psychological problems are predicting NSSI positively, which is consistent with previous research results [1,40]. O’Connor proposed the motivation-will integration model of NSSI occurrence. This model uses stressful life events as the pre-motivation stage of NSSI and adverse emotional or psychopathological symptoms as the motivation stage. The above factors occur during the onset of NSSI, and after, self-injury will apparently be reduced and accompanied by a sense of relief, which gradually promotes the occurrence of NSSI [41]. Nixon believes that NSSI may be a self-treatment mechanism for psychopathological symptoms, especially considering the effects of self-injury on the regulation and addiction characteristics [42]. Scholars through middle school and college student research cohorts also suggest that early psychological stress or psychopathological symptoms will significantly increase the incidence of NSSI in later stage NSSI [40,43].

The bootstrap test shows the indirect impact of school bullying on NSSI through psychological problems of primary and middle school students, indicating that the poor mental health of primary and middle school students can strengthen the emergence of NSSI due to school bullying. Therefore, psychological problems play a mediating role between school bullying and NSSI, which is consistent with the results of Baiden’s study and extensive previous studies that the victims of school bullying only had a significant effect on self-injury [5,44,45]. The self-awareness of rural elementary and middle school students in the developing areas of China has gradually increased during their adolescent years, and their behaviors have tended to be similar to that of adults. Due to intergenerational fostering or foster care, lack of family emotional warmth, doting or negligence, most of them can easily form two extreme personalities. One of the personalities is too outgoing, irritable, and excited, sometimes even unable to control his emotions, and often abusing and bullying others. Another kind of personality is introverted, different from other people’s cognition, indifferent emotions, loneliness, and often pay attention to other people’s views about themselves [46]. Micro-sociology scholars believe that if one side blindly avoids conflict, it will break the interpersonal balance and eventually lead to aggressive behavior [47]. Faced with their own vulnerable characteristics, such as passive endurance of conflicts, victims of bullying will unconsciously “attract” bullies to attack, further enhancing bullying behaviors [21]. Over time, internalization and externalization problems often occur. Victims of bullying have difficulty adjusting emotions and are lonely and nervous. In severe cases, they may experience NSSI or even suicide. The general stress theory model holds that: stressors → negative emotions → dangerous behaviors. According to this model theory, school bullying is a source of stress for rural primary and middle school students in developing areas of China, which will bring tension, depression, anxiety, and dangerous behaviors such as NSSI and aggressive behavior in order to release this negative emotion [45]. The mediation effect model of this study also fully confirms this.

Therefore, appropriate measures should be taken, such as (1) The school shall organize a general survey of incoming students’ mental health status and establish a student’s mental health file in order to screen out groups at high risk of bad mental health, so that the necessary guidance and monitoring measures can be taken effectively and early intervention can be carried out specifically; (2) Schools should conduct mental health courses, hold special lectures on the legal system, set up psychological counseling rooms, and carry out psychological counseling work in a well-rounded way; (3) Schools should strengthen the investigation of school bullying behaviors in rural elementary and middle school student groups, such as installing monitoring in the dead corner areas of school grounds, so that students can lose the place of bullying or fighting, and stop bullying behaviors in a timely manner may prevent rural primary and secondary school students from NSSI. At the same time, in order to strictly prevent and handle school bullying incidents, the school must actively cooperate with the public security and judicial departments to help carry out their work. Besides, the school should create a healthy and harmonious campus environment for students.

### Strengths and Limitations

This study explores the relationship between psychological problems, school bullying, and NSSI, using a small sample of rural primary and middle students in developing areas of China, including the use of school bullying and psychological problems that have become more frequent in recent years in Chinese youth and focusing on the mediating role of psychological problems in the association between school bullying and NSSI behaviors. The contribution of school bullying and psychological problems to NSSI adds to the case for the development of trauma-focused interventions in reducing the risk of NSSI among rural primary and middle school students in developing areas of China.

The limitations of this study are as follows: First, the subject of this study was rural primary and middle school students in Jiangxi Province, a developing area of China. Therefore, in the generalization, interpretation of the research results need to pay attention to this background. Second, the answer attitude is not serious for the questionnaires of this study are relatively large, and the respondent is prone to fatigue due to long-term investigation. Additionally, it may inevitably lead to problems of potential underreporting and biased recall because the questionnaires based on the retrospective self-report of various variables. Third, this study is based on cross-sectional data. Therefore, no causal inference can be drawn about the relationship between some factors in this study and NSSI. Further longitudinal studies are needed to determine the causal relationship between related variables and NSSI.

## 5. Conclusions

This study highlights the importance of the link between school bullying, psychological problems, and NSSI among primary and middle school students in rural China. The results of this study indicate that suffering from school bullying and mild/severe psychological problems are risk factors that lead primary and middle school students to NSSI. In addition, psychological problems mediate between school bullying and NSSI. Further research is needed to clarify the underlying mechanisms of this relationship. The results of the above research indicate that obtaining information on school bullying from primary and middle school students with mental health needs can help identify adolescents who may be at risk for NSSI. The intervention projects of schools and education departments at all levels for rural primary and middle school students in China’s developing areas should focus on monitoring bullying behaviors in schools and paying attention to the mental health of primary and middle school students to ensure that they can devote themselves to a normal life with a healthy mind and body.

## Figures and Tables

**Figure 1 ijerph-17-03371-f001:**
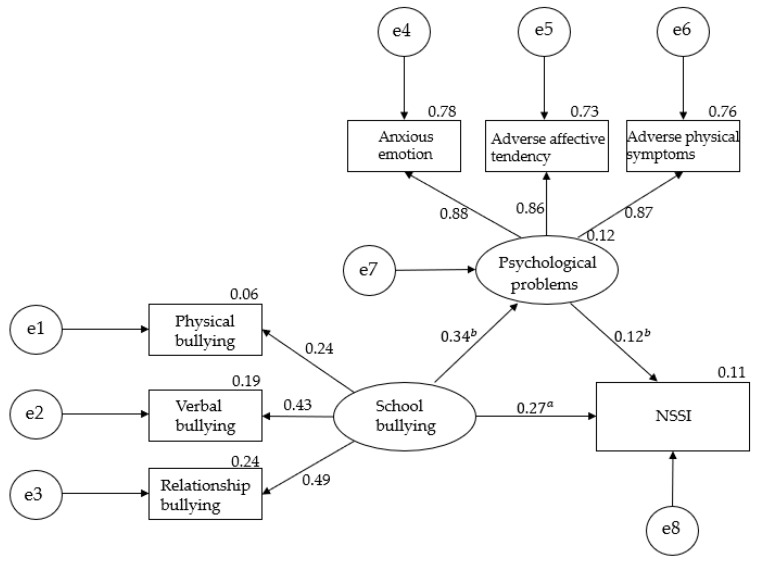
Mediating role of psychological problems in school bullying and NSSI, Note. ^a^
*p* < 0.01; ^b^
*p* < 0.001.

**Table 1 ijerph-17-03371-t001:** Demographic distribution of non-suicidal self-injury behavior among rural primary and middle school students in Jiangxi Province.

Variables	Option	Total *N*	The Number of NSSI (%) *	χ^2^	*P*-Value
Sex					
	Male	1208	186 (15.40)	0.63	>0.05
	Female	1076	153 (14.22)		
Phase of studying					
	Primary school	1348	192 (14.24)	0.93	>0.05
	Middle school	936	147 (15.71)		
Left-behind children					
	Yes	610	92 (15.08)	0.04	>0.05
	No	1674	247 (14.76)		
Only child in one’s family					
	Yes	201	38 (18.91)	2.88	>0.05
	No	2083	301 (14.45)		
Family structure					
	Two-parent family	1904	262 (13.76)	10.60	<0.01
	Non-two-parent family	380	77 (20.26)		
Witnessing household domestic violence					
	Yes	742	128 (17.25)	5.04	<0.05
	No	1542	211 (13.68)		
Physical abuse					
	Yes	2125	328 (15.44)	8.49	<0.01
	No	159	11 (6.92)		
Academic performance					
	High	270	33 (12.22)	1.66	>0.05
	Low/Medium	2014	306 (15.19)		
Academic pressure					
	High	851	158 (18.57)	14.88	<0.01
	Low/Medium	1433	181 (12.63)		
School bullying (past year)					
	Yes	730	176 (24.11)	72.90	<0.01
	No	1554	163 (10.49)		
Psychological problems					
	Normal	587	1 (0.17)	134.56	<0.01
	Mild/Severe	1697	338 (19.92)		

* (%) is the incidence.

**Table 2 ijerph-17-03371-t002:** Influence of school bullying and psychological problems on non-suicidal self-injury (NSSI).

Variable	AOR ^a^ (95% CI)	*P*-Value
School bullying		
Yes	2.10 (1.65, 2.69)	<0.001
No (ref)	1.00	
Psychological problems		
Normal (ref)	1.00	
Mild/Severe	125.03 (17.49, 893.89)	<0.001

Note. AOR = adjusted odds ratio; CI = confidence interval; ref = reference; ^a^ Adjusted for sex, phase of studying, left-behind children, only child in one’s family, family structure, witnessing household domestic violence, physical abuse, academic performance, academic pressure, school bullying, psychological problems.

**Table 3 ijerph-17-03371-t003:** Correlation analysis between factors of school bullying, psychological problems, and NSSI in rural primary and middle school students in Jiangxi Province.

Variable	Physical Bullying	Verbal Bullying	Relationship Bullying	Anxiety	Bad Emotional Tendency	Bad Physical Symptoms
**r**	0.059	0.123	0.164	0.167	0.205	0.177
***P*** **Value**	<0.01	<0.01	<0.01	<0.01	<0.01	<0.01

**Table 4 ijerph-17-03371-t004:** Bootstrap test of significance of one-factor mediation effect.

	Path	95% CI Value		Effect Size
		Lower Limit	Upper Limit	
Direct effect	Bullied at school →NSSI	0.167	0.372	0.267
Mesomeric effect	Bullied at school →Psychological problems →NSSI	0.026	0.056	0.040
Total effect		0.694	1.472	0.307

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
