# Peer review of "Research on the Relationships between Psychological Problems and School Bullying and Non-Suicidal Self-Injury among Rural Primary and Middle School Students in Developing Areas of China"

_ijerph, 2020, doi:10.3390/ijerph17103371_

Round 1

Reviewer 1 Report

Introduction:

Would benefit from a definition of what ‘left-behind youth’ and ‘non-left-behind youth’ is. This is helpful for an international audience.

Reference is made to the incidences of school bullying in primary and secondary school to be 33.36%. Comparing this with other country statistics would be useful. Is this number about average or does China have a lower/higher rate of bullying in comparison to other countries?

The literature would benefit from further exploration on this topic. Currently studies are mentioned but not critiqued, so more detail would be helpful. In addition, other concepts relating to NSSI and bullying at school could be explored such as school safety. A Japanese study by Hamada et al. (2018), found that students (particularly females) who felt unsafe at secondary school, were more likely to engage in self-cutting. However, if the female victims perceived their school as a safe place then self-cutting was less likely.

Reference at the end of the literature review is made to learning “more about Jiangxi's current health policy, preventive screening and education strategies, and develop targeted treatment options.” I suggest telling the reader about this current situation and where the learning could be.

Methods:

Participants and procedure: More detail required here. What types of schools were targeted – privately/publicly funded/ boarding schools etc.? What type pf profile did these schools have e.g. were they typical of all schools in the region? What was the age-range of the participants (for international audience)? Again, a definition of ‘left-behind children’ would be helpful.

Measures: Was a definition of bullying provided for participants? If so was ‘repetition’ included in the definition?

Ethical Aspects: What other ethical implications were at play here? Given the sensitive nature of this study, how was NSSI explained to the participants? As the survey was administered during normal class time, what options were provided to those who did not want to participate?

Discussion:

Some statements are made which either need to be rephrased or supported further by evidence in the literature. For example, the statement that single parents or restructured families tend to be associated with emotional disputes and violent bursts - is this entirely true?

Overall, the discussion is weak in relation to the hypothesis proposed earlier on. There needs to be more about bullying per se and the link to NSSI as uncovered in your study. It would be helpful to review some previous studies in this area in the earlier literature and also to bring them into this discussion. Issues of school safety/school climate could be discussed and intervention programmes focussing on bullying and mental health would add to this discussion overall.

Strengths & Limitations: This sentence is unclear: “It is identified that people with mild/severe psychological problems and those who have experienced school bullying are important people in the prevention of non-suicidal self-injury.” Does this refer to peer support for example?

Limitations on methods such as lengthy questionnaires should be discussed, including how this relates to fatigue/disinterest in children and young people.

Author Response

Please see the attatchment.

Reviewer 2 Report

The article presents research on the relationship between psychological problems, bullying, and the NSSI, using a small sample of rural primary and secondary students in less developed areas of China. Results suggest that those who have experienced bullying and have mild / severe psychological problems are more likely to have NSSI behaviors.

The introduction provides sufficient background on the topic and previews major points. Both research design and analysis are adequate.

A more detailed description of the measuring instruments used in the research is needed. Have they been validated by factorial analysis? If not, why not? This issue is crucial in order to confirm the validity of the data obtained.

It is necessary to review the writing and grammatical style of the document.

The references must be adapted to the format established by the journal.

Round 2

Reviewer 1 Report

Thank you for addressing the comments made which are sufficient. However the paper still requires extensive editing to ensure the English is sufficient and that your points are not lost in translation. 
